# Graphene-Based Materials as Efficient Photocatalysts for Water Splitting

**DOI:** 10.3390/molecules24050906

**Published:** 2019-03-05

**Authors:** Josep Albero, Diego Mateo, Hermenegildo García

**Affiliations:** Instituto Universitario de Tecnología Química CSIC-UPV (ITQ), Avda. de los Naranjos s/n, 46022 Valencia, Spain; diemama@itq.upv.es

**Keywords:** defective graphene, photocatalysis, solar fuels, hydrogen generation, facet-oriented nanoparticles

## Abstract

Photocatalysis has been proposed as one of the most promising approaches for solar fuel production. Among the photocatalysts studied for water splitting, graphene and related materials have recently emerged as attractive candidates due to their striking properties and sustainable production when obtained from biomass wastes. In most of the cases reported so far, graphene has been typically used as additive to enhance its photocatalytic activity of semiconductor materials as consequence of the improved charge separation and visible light harvesting. However, graphene-based materials have demonstrated also intrinsic photocatalytic activity towards solar fuels production, and more specifically for water splitting. The photocatalytic activity of graphene derives from defects generated during synthesis or their introduction through post-synthetic treatments. In this short review, we aim to summarize the most representative examples of graphene based photocatalysts and the different approaches carried out in order to improve the photocatalytic activity towards water splitting. It will be presented that the introduction of defects in the graphenic lattice as well as the incorporation of small amounts of metal or metal oxide nanoparticles on the graphene surface improve the photocatalytic activity of graphene. What is more, a simple one-step preparation method has demonstrated to provide crystal orientation to the nanoparticles strongly grafted on graphene resulting in remarkable photocatalytic properties. These two features, crystal orientation and strong grafting, have been identified as a general methodology to further enhance the photocatalytic activity in graphenebased materials for water splitting. Finally, future prospects in this filed will be also commented.

## 1. Introduction

The massive consumption of fossil fuels that characterizes the modern society has led to a fast increasingly depletion of the resources as well as serious negative environmental impacts related to global warming and climate change. For all those reasons, new approaches for the fabrication of environmentally friendly and cheap fuels whose production is based on renewable energy sources is attracting the interest not only of scientific community but also the industry. Fuel production using solar light as primary energy, also called “solar fuels”, has been proposed in the last decades as one of the most promising approaches. Among them, the photocatalytic water splitting reaction for the production of H_2_ as a potential energy vector is a straightforward process that could make H_2_ available from H_2_O [1].

H_2_ has been typically obtained from fossil fuels and natural gas reforming [2]. Although the process has reached high maturity and efficiency it is not sustainable and lead to CO_2_ emission. On the contrary, photocatalytic water splitting using solar light could be a very suitable method for H_2_ production since it aims at the conversion of solar energy into H_2_ from water. The splitting of water in H_2_ and O_2_ by UV light and electrochemical bias was first reported in 1972 by Fujishima and Honda using TiO_2_ photoelectrodes [3]. In photocatalytic water splitting, oxidation and reduction reactions take place simultaneously in a photocatalysts while this is irradiated. The most typical photocatalysts are based on inorganic semiconductor materials. The photocatalytic water splitting mechanism is based in 4 main steps as depicted in Scheme 1 (elementary steps i to iv). The first step consists in the photogeneration of electron-hole pairs in the semiconductor upon light absorption. The second step involves the migration of the photogenerated electrons and holes through the photocatalyst reaching the external surface. Subsequent steps consist in the oxidation and reduction reactions by the electrons and holes that become in contact with water. Finally, undesirable charge recombination may take place during electron and holes migration or at the photocatalyst surface resulting in charge annihilation.

Among the most studied photocatalysts for water splitting, transition metal oxides, such as TiO_2_, ZnO, WO_3_, metal chalcogenides (CdS, CdSe, CdTe, etc.) or inorganic perovskites (SrTiO_3_, etc.), among others, have been the most intensively studied [4]. However, the still unsatisfactory efficiency makes necessary to improve the performance of the photocatalysts and to find suitable alternatives. In this regard, a new area that is attracting much interest in the last years is the use of graphene (G) or G-based materials as additive or even as active photocatalyst for solar light-driven fuels production [5,6,7]. Gs, when prepared from biomass, are considered renewable materials and their use can contribute to the sustainability of the chemical industry compared to the metal-based photocatalysts. 

Among the properties that justify the interest of G beyond microelectronics, the most important ones are its large surface area, the strong adsorption capacity, possibility to provide charge density at certain stages of the reaction, rational design of active sites by modification of the G sheet during or after the synthesis, and as it will be shown later, the strong interaction between transition metals and the Gs as consequence of the overlapping between the d orbitals in metals and the π orbitals of Gs [8,9].

Contrary to initial studies, where the role of G in photocatalysis was limited to assist and enhance the photocatalytic activity of semiconductor materials by active as co-catalysts, it was demonstrated later that some G-based materials possess intrinsic photocatalytic activity [10]. This behavior is due to the presence of defects on G, either as consequence of the precursor in the preparation, their purposed generation in the graphenic lattice during synthesis or by using post-synthetic procedures. The most common defects in G have been identified as heteroatoms (O, N, B, P, ect.) substituting C atoms, carbon vacancies and lattice holes derived from CO_2_ and CO evolution during G formation [11]. In contrast to ideal G, which has been defined as a 0 band gap semiconductor, the presence of these defects in the lattice originates the semiconductor behavior in these materials. Moreover, it has been demonstrated that the presence of small amounts of metal or metal oxide nanoparticles (NPs) in close contact with the G sheets forms an heterojunction that can improve its photocatalytic activity [12]. The most widely accepted mechanism to explain the improvement in the photocatalytic activity of these hybrid systems is related to charge transfer from the semiconductor conduction band minimum (CBM) to G. The high electron mobility in G determines that, once the electron is on G, they become delocalized through the sheet. In that way, the close contact between the metals or metal oxides and G favors photoinduced charge separation. Scheme 2 represents the most accepted mechanism for the promotion of the photoinduced charge separation between metal or metal oxides and G. 

However, despite the addition of small amounts of metal and metal oxide NPs on Gs is being widely explored for photocatalysis, less attention has been paid to the influence that preparation of NPs supported on Gs and the preferential facet exposure can play in order to further improve their photocatalytic activity. In this regard, it has been widely reported that, for instance, the photocatalytic activity of the 001 facet in anatase TiO_2_ crystals is significantly higher than that of the usually dominating 101 facet, and as a consequence, most studies have focused on the growth of anatase NPs with customized crystallographic structure [13,14]. In a similar way, it has been recently demonstrated that the metal or metal oxide NPs supported on G materials can be preferentially oriented in one of their crystallographic facet, providing a notable enhancement in their catalytic activity.

In this review we will present recent advances in the use of G materials as photocatalysts for water splitting. In addition, it will be show that small amounts of NPs supported on G may improve significantly their photocatalytic activity particularly for water splitting when they exhibit crystal orientation. Our aim is not to exhaustively cover the existing literature on G materials in photocatalysts, but instead to offer an overview of some promising approaches to increase further their intrinsic photocatalytic activity. 

## 2. G-Based Materials. Properties and Preparation

G is a one atom thick 2D material of an indefinite number of sp^2^-hybridized carbon atoms in hexagonal arrangement. The thickness of isolated G sheets is 0.34 nm, accordingly, G is the physical limit in the miniaturization of a 2D material. G is the structural constitutive unit of graphite, which is obtained by strictly ordered stacking in the vertical direction of G layers. The crystal structure of graphite and the stacking derives from the strong π-π interactions between the G layers. Scheme 3 shows the relationship between G and the structure of graphite. 

G and G-based materials have attracted considerable interest in recent years due to their unique mechanical, physical and chemical properties [15]. In contrast to graphite, G is extremely robust, flexible and elastic, having a Young’s modulus several orders of magnitude higher than steel. Although G has been considered a “flat surface”, G layers can bent showing wrinkles and a more corrugated morphology (see Figure 1). The extended π orbital of G has electron density above and below the basal plane. This extended conjugation is responsible for the high-electrical and thermal conductivity of G as well as the high adsorption capacity and the strong interaction with transition metals as commented later. Moreover, G morphology provides a large specific surface area. Theoretical calculations estimate that single layer G should have a specific surface area of approximately 2650 m^2^/g [16,17], which is largest than, for instance, most zeolites and other inorganic porous materials. As consequence of these properties G is considered as a material having totally accessible sites on large surface area with exposed π orbitals free to interact with substrates. All these features are very convenient for catalytic applications.

Another remarkable property of G for the present paper is its transparency to the light in almost the whole UV-Vis spectral range [15]. Single layer G transmittance should be above 99.8%. However, transmittance of two layers is reduced to ≈ 95% and subsequent layer stacking reduce the transmittance to 90% and 70% for three and four layers, respectively. In addition of layer stacking defects and doping converting G into a semiconducting material introduces electronic transition bands in the UV zone that can extend into the visible region, that can be responsible for photocatalytic activation of defective Gs.

Figure 1 shows the typical absorption spectrum of a few layers defective G film obtained from the pyrolysis of a suitable precursor and a high resolution transmission electron microscopy (HRTEM) image of a G sheet.

As can be observed in Figure 1, the absorption spectrum extends from the UV into the visible towards the NIR, overlapping the solar spectrum in the UV-Vis region, being a good candidate as solar light photocatalyst.

The combination of totally accessible high surface area and strong substrate interaction due to exposed π orbitals, with sustainability when obtained from biomass wastes makes G-based materials very attractive for different applications including photocatalysis [18]. For these reasons, the use of G-based materials, either additive or as photoactive component for the photocatalytic production of solar fuels, is a very active research area.

### 2.1. G-Based Materials Preparation Procedures

Single-layer G was first isolated by graphite peel-off using Scotch adhesive tape and deposition onto a silicon substrate [19]. However, among the most widely employed techniques for ideal G film preparation, meaning G films containing a low defect density, the synthesis based on chemical vapor deposition (CVD) using metal surfaces as template has been the most preferred one [19,20]. In this method, the G precursor, typically a hydrocarbon such as CH_4_, is contacted under a H_2_ atmosphere at high temperature (≈ 1500 ^O^C) with the surface of a dehydrogenating metal. The metallic surface is preferably constituted by clean Ni or Cu metals with exposed 100 facets. It is supposed that CH_4_ under these conditions of pressure and temperature undergoes, in the presence of these metals, decomposition giving rise to H_2_ and C atoms that deposit on the metallic surface. It has been demonstrated that the C atoms are deposited surrounding the metal atoms, whose spherical shape and size perfectly fits in the 6 C ring characteristic of G. The crystallinity of the metal film is playing a crucial role templating the growth of the G sheet. Subsequent expansion of the G domains results in the formation of a single G sheets on the surface of the metal template. CVD synthesis provides a high quality, electronic grade G containing low defect concentration. This type of G films has been widely employed in microelectronics due to the high electrical conductivity, electron mobility and carrier density. However, CVD is not the best-suited technique for large scale production of affordable photocatalyst due to the slow formation rate and the need of dedicated equipment. In addition, the lack of defects makes these G devoid of photocatalytic activity.

Apart from G films, many other applications require the presence of G materials suspended in liquids forming inks. These inks cannot be obtained by direct exfoliation of graphite, since this material is composed by the stacking in perfect alignment of G sheets maximizing the π-π interactions. This strong interaction makes extremely difficult the exfoliation of graphite by direct physical methods. Ultrasound treatment is the most common process to attempt exfoliation of graphite is highly-viscous, high-boiling point solvents, but the yield of the process in terms of percentage of graphite mass that undergoes exfoliation is low [21,22]. Scheme 4 shows the different procedures for ideal G fabrication.

Nevertheless, there are plenty of G types far from ideal G obtained from different methods. While ideal Gs can find application in microelectronics, for catalysis defective Gs are much more adequate since the active sites are generally associated to the presence of defects.

Defects in G can be carbon vacancies in the layer. If several continuous carbon atoms are missing, a hole can appear on the G sheet. These holes contain at the periphery carbon atoms that may have incomplete valence, denoted generally as C atoms having “*dangling”* bonds. Other defects that can be found are five or seven member rings. These rings can introduce local strains on the sheet and can even generate local curvature of the G sheet. Besides C vacancies, holes and no hexagonal rings, the most common defects in G-based materials are related to the presence of heteroatoms, especially oxygenated functional groups at the boundary of the G sheets, such as hydroxyl, carbonyl or carboxyl groups.

G-based materials having different O content can be obtained in a series of consecutive steps from graphite by first monitoring a deep chemical oxidation to graphite oxide, followed by a subsequent exfoliation to graphene oxide (GO) and a final partial reduction to reduced GO (r-GO) [23]. r-GO is a highly defective type of G material. The key point in the preparation of r-GO is the transformation of graphite in graphite oxide, since the interaction between GO sheets in graphite oxide is much weaker than the interaction between the G sheets in graphite, and, therefore, graphite oxide is much easier to exfoliate. A very convenient chemical oxidation of graphite was developed by Hummers and Offenbach [24]. Graphite oxide can be easily exfoliated in low viscosity solvents such as water, obtaining persistent suspensions of GO due to its high hydrophobicity. Depending on the chemical oxidation conditions GO oxygen content can vary from 40 to 60 wt%. Finally, GO can be easily reduced by physical or chemical methods. The resulting r-GO has a considerably high density of defects compared to G obtained by CVD as consequence of the presence of residual oxygen (10–20 wt%) and holes from evolved CO_2_. However, GO and r-GO have been found very suitable photocatalysts for water splitting as it will be commented in the following sections. Scheme 5 summarizes the sequential steps involving GO and r-GO preparation from graphite. 

Besides the introduction of O, other elements different from C have been found also very interesting as dopants for the preparation of G-based materials as photocatalysts are N, S, B and P. A typical preparation procedure of these doped-Gs starts with GO that reacts with a suitable reagent containing the dopant element, for instance NH_3_ or H_2_S. Epoxy, hydroxyl and carbonyl groups present in GO can react with the reactive molecule of the dopant element by substitution, condensation or nucleophilic attack leading to a doped r-GO. Reduction with partial reconstitution of the G structure occurs generally during the process of heteroatom incorporation. 

Another procedure for the preparation of doped-Gs that has been reported by our group consists in the pyrolysis under inert atmosphere of natural biopolymers or convenient derivatives [25]. For instance, chitosan, a polysaccharide of glucosamine can act simultaneously as N and C source, rendering N-doped G with a 5 wt% of N and 8 wt% of O in the composition depending on the pyrolysis conditions. Introduction of other elements not present in the polysaccharide composition is also possible and, for instance, sodium alginate can be esterified with boric or phosphoric acids as consequence of the tendency of hydroxyl groups of saccharides to form esters with inorganic acids. Pyrolysis of these polysaccharides esters of inorganic acids result in formation of doped G containing the heteroatoms initially present in the inorganic moiety of the ester as well as other heteroatoms already existing in the polysaccharide. This may result in the formation of co-doped Gs, for example N and B co-doped G by pyrolysis of the borate esters of chitosan. This pyrolytic process can be applied either to the preparation of G films on arbitrary substrates or to the preparation of G suspensions in different solvents. Scheme 5 presents a summary of the different preparation procedures employed for the preparation of some G-based materials.

### 2.2. G as Support of Metal or Metal Oxide NPs

G-based materials are very suitable as supports of metal or metal oxide NPs as previously commented. The main characteristics of G as support of metal or metal oxide NPs are its large surface area and the strong metal-G interaction, which can lead to stabilization of the supported NPs, immobilizing them on the surface and avoiding their leaching or growth [18]. This strong interaction derives from the overlap of the extended π orbitals of G with the atomic d orbitals of the metal NP. This overlap can also tune the electronic density of the interfacial metal atoms by partial charge transfer between the G support and the metal. The presence of defects such as vacancies, dangling bonds and dopant elements on G has been found to play an important role for the interaction with metallic NPs too. Thus, theoretical calculations have revealed that the interaction energy between Au and other metal NPs and N-doped G should be higher than in the case of non-doped G as consequence of the stronger interaction between Au and N atoms compared to C atoms. In contrast, other types of metals such as Cr and Fe can form metal carbides as consequence of their higher affinity for carbon [26,27]. 

The most common method used for NPs deposition in G-based materials is reminiscent of the so-called deposition-precipitation method described by Haruta for inorganic oxides [28,29], where the formation of the NPs-G occurs via two different steps. In a first step, G-based materials are suspended in water containing the desired metal salt at basic pH. Metal ions adsorbed chemically to the G-based material undergoing spontaneous agglomeration due to their strong interaction. In this step, metal ions associate with the G-based material by a combination of Coulombic forces, covalent interactions and van der Walls forces. The second step involves the reduction of the metal ions on the surface. Thus, reduction step is typically carried out under a H_2_ flow at mild temperatures (100–400 °C) or by contacting the solid with a reducing agent (alcohols, tertiary amines, etc.) in the liquid phase. A modification of this method has been named “impregnation method” and it is carried out by adsorption of the metal ions on the G surface by slow solvent evaporation of the aqueous suspensions containing the metal salt and the G-based material. The obtained dry solid is subsequently reduced using a H_2_ steam at moderate temperatures.

A similar approach can be carried out without the presence of water during deposition but using instead a high boiling point, viscous alcohol such as ethylene glycol, accosting so-called “solvothermal method” [30]. The process involves heating the G material dispersed in a solution of the metal salt and the ethylene glycol at moderate temperatures (80–160 °C). In this case, the ethylene glycol acts as solvent and reducing agent of the metal salt that becomes converted to metal NP and adsorbs on the G surface.

In a related strategy, preformed NPs are deposited on the G materials. In this methodology, first NPs are synthetized utilizing capping ligands in order to stabilize the NPs. Then, these metal NPs in suspension are adsorbed on the G surface by dispersing the two materials in a common solvent. Subsequent thermal annealing is sometimes carried out in order to further enhance the interaction between the NPs and G [31,32].

Another possibility to deposit metal NPs on G is based on photochemical reduction. In this method a G material is irradiated in the presence of a sacrificial electron donor. In this case the G-based material is acting as photocatalyst, and under illumination photo-induced charge separation takes place, generating electron and holes on G. In the presence of alcohols or tertiary amines as sacrificial agents, the holes are quenched by these chemicals leaving the electrons that reduce the metal ions to NPs on the G surface [33]. Scheme 6 summarizes the different methods employed to support NPs on G-based materials. Metal oxide NPs can be easily obtained by spontaneous oxidation of metal NPs upon exposure to the air.

## 3. G-Based Materials as Photocatalysts for Water Splitting 

G is a 0 band gap semiconductor due to the electronic band overlap. This conductive property of G impedes its application in photocatalysis since electrons and holes would undergo instantaneous recombination [34]. Therefore, band gap opening by doping with heteroatoms or/and by introducing lattice defects is necessary to convert metallic ideal G into a semiconductor defective G material. In contrast to ideal G, GO and r-GO, both containing oxygen functionalities, behave as semiconductor materials. It is well known that the introduction of small amount of oxygenated functional groups transforms a portion of the characteristic G sp^2^ carbons into sp^3^. Further increase in the oxygen content promotes the co-existence of conductive sp^2^ and non-conductive sp^3^ domains leading to the opening of a band gap. GO behaves as a p-type semiconductor due to its lower electron mobility. The GO band gap has been determined to be between 2.4 and 4.3 eV [35], depending on the oxidation level. An increase in oxygen content promotes band gap enlargement since the valence band maximum (VBM) gradually changes from the G π orbital to the oxygen 2p orbital, while conduction band minimum (CBM) remains unchanged in the G π* orbital [36]. 

Thus, Yeh et al. reported the photocatalytic activity of GO for water splitting obtaining a H_2_ production of 2833 µmol/h in the presence of CH_3_OH as sacrificial electron donor and 47 µmol/h in the absence of sacrificial agents [35]. Later, the same authors prepared different GO photocatalysts with different oxygen content and studied the photocatalytic activity towards water splitting as function of the GO oxidation degree [37]. It was demonstrated that a gradual increase in oxygen produced a decrease in the GO VBM. Thus, the GO with the highest oxidation potential was able to produce O_2_ while the less oxygenated GO materials did not. The authors observed that only the GO containing the maximum oxygen content had sufficient positive VBM potential for water oxidation, although H_2_ evolution was observed in all cases. However, the highly oxidized GO presented a large band gap, sluffing light absorption towards deep UV and thus, obtaining lower H_2_ production than the GO with the lower oxidation level but the smallest band gap [37]. Scheme 7 shows the different VBM and CBM for the different GO as a function of the oxygen content in connection with the water reduction and oxidation potentials.

For inorganic semiconductor oxides, doping with non-metal elements have been widely reported in photocatalysis in order to enhance light harvesting, especially in the visible region of the solar spectrum, where typical wide band gap semiconductors cannot absorb photons. In the case of TiO_2_ it has been demonstrated that photocatalyst doping with elements, such as N or S, can introduce energy levels of higher energy than the semiconductor VBM, producing a band gap shortening [38,39,40]. In the G case, substitution of C atoms by other heteroatoms (N, B, P, etc.) has also demonstrated to influence the charge density distribution on G, modifying the band alignment as well as the optical and chemical properties. The heteroatoms can exhibit preferred tendency to act as electron donor or acceptor, depending on their electronegativity with respect to C. The most widely explored dopant element in G-based materials has been N, although other elements have been studied too. In this way, Latorre-Sánchez et al. prepared N and P doped r-GO by pyrolysis at 900 °C in Ar atmosphere of chitosan and H_3_PO_4_-modified alginate, respectively [41,42]. Chitosan pyrolysis has been reported to produce a turbostratic carbon residue that after exfoliation gives rise to a defective G material with oxygen and nitrogen contents of approximately 10 and 5 wt%., respectively [25]. In a similar way, the pyrolysis of the phosphate modified ester obtained from sodium alginate affords defective G with oxygen and phosphorous content below 10 and around 1.5 wt%, respectively, after exfoliation of the carbon residue. The as-prepared P doped defective G produced H_2_ from H_2_O: CH_3_OH mixture at a rate of 12 µmol/g·h under UV-Vis light irradiation with a Xe lamp. This production rate was found 10 times higher than the obtained from irradiation irradiation of a r-GO analog obtained from sodium alginate pyrolysis lacking P doping [41]. On the other hand, the photocatalytic activity of N doped defective G obtained from the pyrolysis of chitosan was of 18 µmol/h upon irradiation of a H_2_O: CH_3_OH mixture in the visible region with a 532 nm laser [42]. 

In a similar way, Chai and coworkers have reported recently the preparation of N-doped r-GO and B-doped r-GO obtained by pyrolysis of GO in the presence of urea and boron anhydride, respectively [43]. The N and B contents were found of 8.26 and 3.59 at%, respectively. The photocatalytic activity of these doped G materials in 0.1 M Na_2_S/Na_2_SO_3_ aqueous solutions in the visible region was of approximately 65 µmol/g·h for both photocatalysts, which was found 2.6 and 5.3 times higher than the H_2_ production rate measured for undoped GO and r-GO, respectively [43]. 

It is worth noticing that G-based materials can be doped by one, two or even more heteroatoms. In an elegant example, N-, P- and F-doped GO electrocatalyst was obtained upon thermal treatment of polyaniline coated GO and NH_4_PF_6_ [44]. The aim of this multi-doped G material was to generate domains with opposite n or p semiconductor behavior to promote simultaneously water reduction and oxidation reactions over each individual sheet. The N-, P- and F-doped GO produced, electrocatalytically in pure water, H_2_ and O_2_ rates of 0.496 and 0.254 µL/s, respectively [44].

N-doped GO quantum dots (QDs) with lateral dimensions of less than 100 nm have been found also suitable photocatalysts for water splitting. Electrochemical impedance spectroscopy revealed that these N-doped GO QDs presented both p- and n- type semiconductor behavior. Thus, both n- and p- domains work as interfacial junction where charge separation can take place. The band gap of the N doped-GO QDs was found 2.2 eV, and under visible light irradiation (λ > 420 nm) in the solely presence of water, stoichiometric amounts of H_2_ and O_2_ were detected [45]. Scheme 8 shows the reaction mechanism in the N-doped GO QDs with different p- and n- type semiconductor domains.

In a similar approach, S-doped GO QDs were prepared by a hydrothermal method using citric acid as GO QDs precursor and NaHS as dopant source [46]. The aim of the S doping was to enhance light harvesting of the GO QDs in the visible region. The authors determined the GO and S-doped GO QD band gap obtaining values of 2.8 and 2.3 eV, respectively. The UV-Vis absorption spectrum of the S-doped GO QDs presented three absorption bands with maximum at 333, 395 and 524 nm, respectively. S-doped GO QDs exhibit at pH 8, and 95 °C a H_2_ production rate of 303 µmol/g·h. However, when 20% (v/v) of ethanol was included in the reaction mixture at the same conditions 508 µmol/g·h were measured. Further increase in the alcohol concentration produced an improvement of the photocatalytic activity up to reaching 1471 µmol/g·h of H_2_ when 80% (v/v) ethanol was used [46]. 

Overall, the introduction of lattice defects on the G sheets by doping with heteroatoms has demonstrated to be a very convenient method for opening the G bandgap and, thus, for converting metallic ideal G into semiconductor materials with excellent photocatalytic properties for water splitting. Table 1 summarizes the H_2_ production and reaction conditions of some reported G-based photocatalysts.

## 4. Metal and Metal Oxide NPs Supported on G-Based Materials as Photocatalysts

### 4.1. Randomly Oriented Metal or Metal Oxide NPs

The characteristic 0 band gap and excellent electronic properties exhibited by G has motivated in the last years the use of this and related materials in hybrid nanocomposites with metal NPs (Ni, Au, Pt, etc.) or inorganic semiconductors (TiO_2_, ZnO, CdS, BiVO_4_, etc) in photocatalytic applications [10,47]. Since Kamat and co-worker reported the preparation of TiO_2_/G nanocomposites by GO photoreduction [48], a great number of examples have appeared in the literature where G-based materials have used as additives of wide bad gap semiconductors, acting either as electron acceptor or/and sensitizer [49]. For instance, the heterojunction between GO (as p-type semiconductor) and TiO_2_ (as n-type semiconductor) has been found to enhance charge separation yield and extend the electron lifetime. TiO_2_ is a well-known wide band gap semiconductor whose light absorption is limited to the UV region being charge recombination very fast [50,51]. The presence of GO has been found very convenient to enhance charge separation in TiO_2_ due to high interfacial contact between TiO_2_ and 2D GO. In addition, the interaction of GO electrons in the π orbitals with Ti atoms in the TiO_2_ surface extends the light harvesting of the latter inorganic oxide into the visible range [52].

GO has been also used as electron transport mediator to facilitate charge separation in Z-scheme photocatalysts. Iwase et al. developed a system where GO was deposited on BiVO_4_, as efficient O_2_ evolution photocatalyst. Subsequently, the rGO-BiVO_4_ was mixed with Ru-SrTiO_3_:Rh, as H_2_ evolution photocatalyst, forming a composite where rGO act as redox mediator transferring photogenerated electrons from BiVO_4_ CBM to Ru-SrTiO_3_:Rh VBM [53]. As consequence, the remaining photogenerated electrons in Ru-SrTiO_3_:Rh CBM and holes in rGO-BiVO_4_ VBM reacted with water to form H_2_ and O_2_, respectively. Scheme 9 shows the operation of the Z-scheme photocatalyst constituted by rGO-BiVO_4_/Ru-SrTiO_3_:Rh for water splitting.

Since the properties and performance of hybrid composites between G-based materials and other photocatalysts have been reviewed, the reader is referred to the existing references for full coverage of this area [5,6,49,54,55,56]. In the present review, we have focused on the use of metal or metal oxide NPs as cocatalysts of G-based materials photocatalysts for improvement of the photocatalytic activity towards water splitting. Unlike the use of G-based materials as additive in photocatalysis, metal- or metal oxide decorated G materials have been seldom studied, although it can be expected that they will receive more attention due of the low amount of metals and improved activity that these materials may presents. In this regard, Ni and NiO NPs were found very convenient co-catalysts to enhance the photocatalytic H_2_ production under Hg-lamp irradiation in H_2_O:CH_3_OH mixtures [57]. Ni or NiO NPs supported on GO promoted a four-fold and seven-fold improvement, respectively, in photocatalytic H_2_ evolution when compared with the activity of bare GO as photocatalyst. Ni NPs were deposited on GO (Ni-GO) by ion adsorption and chemical reduction using NaBH_4_. Subsequent thermal treatment of the Ni- GO, in air conditions, gave rise to NiO NPs supported on GO (NiO-GO) [57].

Recently, Diaz-Torres and co-workers reported the growth of Ni(OH)_2_ NPs on G films using a photoreduction method [58]. In this study, Ni(NO_3_)_2_ was dissolved in a mixture of ethylene glycol and DMF, subsequently G films were soaked in this solution and irradiated with monochromatic 365 nm UV light. Then, NaBH_4_ was added to the mixture and stirred for 5 h. The Ni(OH)_2_ NPs supported on the G films presented in water a photocatalytic activity for H_2_ evolution upon UV light irradiation (254 nm) of 330 µmol/g, while G film in the absence of Ni(OH)_2_ NPs produce only 90 µmol/g. Interestingly, the presence of O_2_ was not detected in the photocatalytic experiments. Instead, it was observed that the G surface suffered oxidation during the photocatalytic reaction generating oxygenated groups which have been reported to act as active sites in the photogeneration of electron-hole pairs and contribute to the overall H_2_ production [58].

Similarly, r-GO decorated with Ni NPs was prepared by one-pot reduction using GO and a Ni salt as precursors [12]. A synergism between the r-GO and the Ni NPs was found in the photocatalytic H_2_ production. The author found that this synergy arises from the coupling of the Ni NPs and the r-GO defects. At a 6 wt% Ni loading, the photocatalytic activity upon visible light irradiation (λ > 420 nm) of Ni-r-GO in aqueous dispersions containing trimethylamine was of 94 µmol/h, corresponding to a Quantum yield (QY) of 30.3% at 470 nm [12]. 

Carbon QDs have been also used as support of metal NPs. For instance, Chen et al. prepared ammonia modified N-doped GO QDs using a hydrothermal method [59]. The ammonia treatment was found to transform some pyridinic/pyrrolic groups into amino/amide groups in the N-doped GO QDs. Pt NPs were loaded in the modified N-doped GO QDs and upon irradiation this material achieved 21% QY in water:triethanolamine [59]. 

In a different approach, MoS_2_ nanoplatelets were grown on r-GO sheets in one step synthesis through pyrolysis at 900 °C under Ar atmosphere of alginate films or powders containing small amounts of (NH_4_)_2_MoS_4_ [60]. It was demonstrated that under these conditions (NH_4_)_2_MoS_4_ decomposes to MoS_2_ on the r-GO surface, establishing a strong interaction between the two materials. Photocatalytic experiments showed a H_2_ production rate over 1 mmol/g·h upon aqueous dispersion containing 15% of triethanolamine (v:v) using radiation from filtered Xe lamp (λ > 390 nm) at 130 mW/cm^2^ [60]. 

### 4.2. Preferential Orientation in Metal or Metal Oxide NPs

Theoretical calculations and available experimental data have demonstrated that specific crystal facets of NPs can exhibit different catalytic activities [61,62]. In this regard, there is an increasing interest in control the preferential facets of NPs exposed to the reaction media and to determine the effect of this preferential crystallographic orientation on the (photo)catalytic activity of metal or metal oxide NPs. More related with the present work, in the case of overall water splitting, Lee and coworkers reported that the morphology of Cu_2_O NPs strongly influence the photocatalytic activity and stability for H_2_ production from water and visible light irradiation [63]. In that work, the production of H_2_ with NPs of different morphologies (cubes, octahedra and rhombic dodecahedra), exhibiting preferential orientation in the 100, 111 and 110 facets were studied. The different morphologies produced H_2_ production rates of 0, 0.5 and 1.6 µmol/g·h for cubes, octahedral and rhombic dodecahedra, respectively. However, it was observed a fast degradation of the photocatalytic activity as consequence of the photocorrosion of Cu_2_O to CuO during reactions [63]. The influence of the crystal facets of Cu_2_O in the energy bands values and on the stability of CuO_2_ discussed by means DFT calculations [64], concluding that the search of preparation methods of Cu_2_O NPs with preferential orientation would be a powerful tool to enhance the photocatalytic activity of Cu_2_O in water splitting [64]. 

In this regard, our group has developed recently a suitable preparation method of preferentially oriented metal or metal oxide NPs supported on few layer G films deposited on arbitrary substrates [65]. The preparation method is based on thin films of biomass wastes such as chitosan or alginic acid deposited by spin coating. The G precursor films are subsequently soaked in aqueous solutions of the metal or metal oxide NPs salts for a few seconds. The modified G precursor films are pyrolized at 900 °C under an Ar atmosphere, obtaining spontaneously oriented metal NPs supported on few layers G films. The formation of metal NPs with preferential orientation can be explained by the so-called “reverse templating effect”. According to this proposal, contrary to CVD deposition, preferential growth of metal facets would occur as consequence of epitaxial growth of the metal on the G surface. In the reverse templating effect, the G hexagonal arrangement will determine the epitaxial growth of the NPs, leading to the development preferentially of these facets that match the geometry and dimensions of the G structure. Subsequently these metal NPs with preferential orientation would undergo spontaneous oxidation to their corresponding oxides, also with preferential crystal facets, upon exposure to the ambient oxygen.

Different metals or metal oxides NPs supported on few layer G films demonstrating a preferential orientation in one of its crystal facets have been already reported using this method, and they have been applied as catalysts or photocatalysts in different reactions exhibiting always a superior catalytic activity that the corresponding materials without any preferential orientation [65,66,67]. Depending on the nature of metal to be adsorbed on the G precursor the described method can be slightly modified by dissolving the metal salt in the G precursor aqueous solution. It that way, the G precursor can be spin coated on the quartz substrates already containing the metal salt, and therefore, pyrolysis of these composites can be directly carried out without the need to soak films into aqueous metal salts solutions. Scheme 10 illustrates this experimental procedure.

The latter was the case in the preparation of 200 Cu_2_O nanoplatelets supported on few layers G [68]. Interestingly, it was observed that metal Cu NPs oriented in the 111 facet were obtained on the surface of the few layers G films immediately after pyrolysis. However, upon ambient exposure the metal NPs become over the time spontaneously oxidized to Cu_2_O exhibiting preferential orientation to the 200 facets. It is worth noticing that in the oxidation the resulting Cu_2_O nanoplatelets exhibit also a preferential orientation in one of their facets reflecting probably that the precursor Cu NPs were already oriented [68]. The as prepared films were submitted to UV-Vis light irradiation with a 300 W Xe lamp in pure water, and the photocatalytic activity tested. H_2_ and O_2_ in stoichiometric amounts were detected, obtaining a H_2_ production rate of 19.5 mmol/g·h. In contrast, the photocatalytic activity of an analogous sample consisting in randomly oriented Cu_2_O NPs supported of few layers G prepared by physisorption of preformed Cu_2_O NPs on G films at identical loading was found 4 orders of magnitude lower. In this case, the one step pyrolysis preparation procedure was not only responsible of the preferential Cu_2_O 200 facet orientation, but also of the strong grafting of the metal oxide NPs on the G surface which promoted a strong interaction between these two materials favoring the interfacial electron transfer between the two components of the photocatalyst [68]. 

111 oriented Au NPs supported on few layer G were prepared upon chitosan film soaking in aqueous HAuCl_4_ solution and subsequent pyrolysis at 900 °C in Ar atmosphere [69]. Chitosan is a biomass waste obtained from the fishery industry that has demonstrated to be filmogenic, forming crack-free and uniform nanometric coatings on arbitrary substrates [25]. Pyrolysis of this biopolymer has been reported to provide few layer defective G containing different defects, such as the presence of N and O heteroatoms as well as carbon vacancies and holes. Besides, chitosan is well-known to efficiently complex metal ions in aqueous solutions through hydrogen bonding with the amino groups composing the structure of this polysaccharide. For that reason, AuCl_4_^−^ is efficiently adsorbed in the chitosan film upon soaking, and after pyrolysis, Au NPs appear strongly grafted to the G films. A proof of the strong grafting of the Au NPs with the G is the NPs size and morphology. Nanoplatelets with lateral size between 5 and 25 nm have been measured from FESEM images, while nanoplatelet height was 2–4 nm, determined by Atomic Force Microscopy (AFM) [69]. 

The 111 preferential orientation of the Au NPs was determined by different techniques, such as XRD, where only the peaks corresponding to the 111 facet were observed. In addition, another proof of the preferential orientation of the Au NP was obtained by Electron Backscattering Diffraction (EBSD) combined to scanning electron microscopy (SEM) imaging. In EBSD technique, crystallographic orientation of each individual particle in the image can be mapped by scanning the electron beam through the sample to acquire the image, while indexing the resulting diffraction patterns by Energy Dispersive X-Ray (EDX). The comparison of the FESEM image with that obtained by EBCD shows that approximately 90% of the Au NPs present in the image exhibit orientation to the 111 facet. Additionally, HRTEM images in agreement with the 111 facet orientation of the Au NP were obtained by perforating the quartz substrate by initial mechanical polishing, subsequent dimpling grinding and final Ar ion bombardment to acquire the transmission images of the Au-G films. The images showed that the Au NPs exhibit preferential orientation to the 111 facet as indicated by the electron diffraction model with the interplanar distance expected for this facet of 0.23 nm [69]. 

Photocatalytic experiments were carried out in 2 × 2 cm^2^ films using the oriented Au NPs supported on few layer G. The Au content determined by ICP-OES was 1 μg/cm^2^. Stoichiometric amounts of H_2_ and O_2_ were detected upon UV-Vis irradiation of the film with a 300 W Xe lamp in pure water, obtaining a constant H_2_ production rate of approximately 1 mol/g·h over 24 h irradiation [69]. 

Control experiments using unoriented Au NPS supported on few layer G were carried out, obtaining three magnitude orders lower H_2_ production rate than using the oriented NPs. In addition, these oriented Au-G photocatalysts exhibited a notable activity in the visible range (about 20% of the total H_2_ production) due to photon absorption by the Au NPs surface plasmon band that introduces photoresponse in the visible. The remarkable photocatalytic activity of this photocatalyst was attributed to the one-step pyrolytic preparation procedure that causes a strong Au-G grafting and preferential 111 facet orientation of the Au NPs [69].

Using a very similar preparation method, Qu and co-workers have recently reported the photocatalytic activity of 111 Au NPs supported on carbon nitride [70]. In this study, CN films were grown on quartz substrates by vapor deposition polymerization, the films were then soaked in HAuCl_4_ solution and thermally annealed at 200 °C. The 111 oriented Au NPs supported on CN exhibited a H_2_ production rate of 150 µmol/g·h from pure water under visible light irradiation (λ > 420 nm). The authors claimed that the photocatalytic activity of the Au-CN photocatalysts arises from the strong coupling between the 111 oriented Au NPs and the CN, which contributes favorably not only to enhance the photo-induced charge transfer but also to a better band alignment for this reaction [70]. Despite the lower H_2_ production obtained with this photocatalyst, the results on Au-CN indicate that this preparation method of oriented metal NPs can be extrapolated to other 2D materials different from G for the deposition of metal or metal oxide NPs preferentially oriented in one of its facets and exhibiting strong grafting to the substrate. These two characteristics have demonstrated to be crucial to improve the photocatalytic activity for overall water splitting since all the available controls with analogous photocatalysts containing the same components but without any NPs crystal orientation and poorly grafted to the substrate present much poor performance.

Overall, it has been demonstrated that the addition of small amounts of metal or metal oxides onto defective G can greatly improve the photocatalytic activity of G-based photocatalysts towards water splitting. Moreover, it has been determined that the exposure to the reaction media of some preferential facets of the metal or metal oxide co-catalysts as well as a strong grafting between G and the co-catalysts can produce a further increase in the photocatalytic activity compared to analogous photocatalysts lacking this preferential orientation or with loose interaction. Table 1 summarizes the H_2_ production and reaction conditions of the most relevant G-based photocatalysts containing metal or metal oxide NPs as cocatalysts.

**Table 1 molecules-24-00906-t001:** G-based photocatalysts employed for water splitting, H_2_ production and reaction conditions.

Photocatalysts	H_2_ Production	Conditions	Light Source	Reference
GO (O: 28 wt%)	5.67 mmol/g·h	H_2_O/MeOH (80:20, v:v)	400 W Hg lamp	[35]
P/G (C1s /P1s: 12.73 at%)	12 µmol/g·h	H_2_O/MeOH (70:30, v:v)	300 W Xe lamp	[41]
N/G (N: 5.4 wt%)	5 mmol/g·h	H_2_O/MeOH (70:30, v:v)	Laser pulse at 532 nm (1800 mW/cm^2^)	[42]
N/r-GO (8.25 at%)	67 µmol/g·h	0.1M Na_2_S/Na_2_SO_3_ aqueous solution	500 W Xe lamp with visible light cut off	[43]
B/r-GO (3.59 at%)	65 µmol/g·h	0.1M Na_2_S/Na_2_SO_3_ aqueous solution	500 W Xe lamp with visible light cut off	[43]
N/GQDs (N: 6 at%)	0.51 µmol/g·h	Pure H_2_O	300 W Xe lamp with UV light cut off	[45]
S/GQDs (1.9 at%)	568 µmol/g·h	H_2_O/*^i^*PrOH (80:20, v:v), 95 °C, pH 8	500 W Xe lamp	[46]
Ni/GO (Ni: 3 wt%)	70 µmol/ h	H_2_O/MeOH (80:20, v:v)	400 W Hg lamp	[57]
Ni(OH)_2_/G films (Ni(OH)_:_ 1.6 wt%)	110 µmol/g·h	Pure water	Monochromatic 254 nm	[58]
Ni/r-GO (Ni: 8 wt%)	94 µmol/g·h	2·10^−4^ M Eosin Y + 7.7·10^−2^M Trimethylamine aqueous solution	Hg lamp UV light cut off	[12]
MoS_2_/r-GO (MoS_2_: 52.7 wt%)	1.2 mmol/g·h	H_2_O/TEOA (85:15, v:v)	Xe lamp UV light cut off (130 mW/cm^2^	[60]
Oriented CuO_2_/N-G films (CuO_2_: 4.75 µg/cm^2^)	19.5 mmol/g·h	Pure water	300 W Xe lamp	[68]
Oriented Au/N-G films (Au: 1 µg/cm^2^)	1.2 mol/g·h	Pure water	300 W Xe lamp	[69]

## 5. Conclusions and Future Prospects

This account summarizes the photocatalytic activity of G-based materials for water splitting, commenting the main approaches carried out to improve their photocatalytic H_2_ production. Starting from the initial use of graphene and related materials as additives or co-catalysts in semiconductor materials, it has been demonstrated that G-based materials possess intrinsic photocatalytic activity towards water splitting. It has been described that this photocatalytic activity can be improved by selective introduction of defects consisting in heteroatom doping with elements such as O, N, P, B, S etc. in the graphenic lattice. These defects are responsible for opening the band gap in G-based materials, offering also the possibility of additional fine tuning of the band alignment by introducing more than one heteroatom in appropriate extent. When using G as photocatalyst, it has been shown that the photocatalytic activity of these materials can be further improved by addition of small amounts of metal or metal oxide NPs on the G surface as cocatalysts. The experimental evidence obtained so far indicates that a strong grafting between the NPs and the support as well as NPs preferential orientation of one of its crystallographic facets, significantly improves the photocatalytic activity of G-based materials for water splitting. 

The uniqueness of G-based photocatalysts compared to other photocatalytic systems lies, on one hand, in the fine tuning of the optoelectronic properties of the defective G as consequence of heteroatom doping, which is able not only to selectively shift the semiconducting behavior between n- or p- type, depending on the element, but also, the position of the conduction and valence bands in a energy range as function of the doping concentration. Moreover, it has been demonstrated that n/p heterojunctions at the nanoscale can be obtained by preparing films of two Gs doped with two or more different elements. On the other hand, the electronic interaction between metal or metal oxide and G orbitals gives rise to the strong grafting observed between them. This strong interaction benefits efficient photo-induced charge transfer and extended carrier lifetimes, which in overall contribute to the improved photocatalytic activity of these hybrid materials. The improved photocatalytic activity observed in these hybrid photocatalysts is also consequence of the preparation method. The one-step preparation procedure involving G film formation by pyrolysis simultaneously with the formation of metal or metal oxide NPs results in the preferential orientation and the strong grafting with the G substrate. It has been highlighted that this procedure appears as the most promising strategy to obtain highly efficient photocatalysts for water splitting. However, exploration of earth abundant, low cost and non-toxic metal or metal oxide NPs apart from Au that maximize the quantum efficiency for H_2_ production should still be carried out. These G based photocatalysts should exhibit optimum redox potentials, fast charge separation kinetics as well as a proper light harvesting properties in the complete UV-Vis region. In this sense, the combination of two or more metal or metal oxide NPs as well as the preparation of metal alloys supported on G-based materials should be worth to be explored in order to meet several of these requirements. On the other hand, the search for modified preparation procedures that allow the formation at will of a wanted crystal facet would be highly important in order to selectively study the influence of each crystal facet in the photocatalytic activity. The combination of metal or metal oxide NPs strongly grafted on G materials together with the fine tune the crystal facets of the co-catalysts could give rise to a new generation of photocatalysts exhibiting exceptional photocatalytic activity for water splitting, achieving higher quantum yields close to those needed for commercial application.

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
