# Peer review of "Graphene-Based Materials as Efficient Photocatalysts for Water Splitting"

_molecules, 2019, doi:10.3390/molecules24050906_

Round 1

Reviewer 1 Report

The paper is a nice review on the use of graphene-based materials for water splitting.

It is well organized and I recommend acceptance after editing the following:

- More attention has to be given to the comparative performance of graphene-modified photocatalysts and graphene-free photocatalysts, possible introducing a table/figure in the manuscript with the most relevant results in terms of hydrogen evolution.

- The recent review by L.Y. Ozer et al., JPhotochemPhotobiolC, 2017, 33, 132-164, should be cited. It reports on the use of composites made of inorganic semiconductors and graphene for photo(electro)catalytic applications, including hydrogen production.

Author Response

Referee 1:

The paper is a nice review on the use of graphene-based materials for water splitting.

It is well organized and I recommend acceptance after editing the following:

- More attention has to be given to the comparative performance of graphene-modified photocatalysts and graphene-free photocatalysts, possible introducing a table/figure in the manuscript with the most relevant results in terms of hydrogen evolution.

A new Table (Table 1) has been introduced in page 11 of the revised manuscript. This table contains some of the most relevant examples of defective G-based photocatalysts for H2 evolution and showing the reaction conditions.

- The recent review by L.Y. Ozer et al., JPhotochemPhotobiolC, 2017, 33, 132-164, should be cited. It reports on the use of composites made of inorganic semiconductors and graphene for photo(electro)catalytic applications, including hydrogen production.

This review has been cited as ref. 49

Reviewer 2 Report

The authors provided a concise but thorough summary of the titled field. I believe that a comparison of benchmark G-based systems and other representative catalytic systems, including reaction condition, additive(s), catalytic efficiency, etc., will be valuable for the topic. Moreover, a summary/outlook of unique catalytic mechanisms in G-based systems compared to other catalytic systems will significantly benefit readers as well.  

Author Response

The authors provided a concise but thorough summary of the titled field. I believe that a comparison of benchmark G-based systems and other representative catalytic systems, including reaction condition, additive(s), catalytic efficiency, etc., will be valuable for the topic. Moreover, a summary/outlook of unique catalytic mechanisms in G-based systems compared to other catalytic systems will significantly benefit readers as well.  

A summary of the most representative photocatalytic systems has now been included in Table 1. This table contains H2production rates, reaction conditions and light source.

A summary of the unique catalytic properties of graphene based materials has been included in section 5: conclusions and future prospects.